# Effects of IL-6 Signaling Pathway Inhibition on Weight and BMI: A Systematic Review and Meta-Analysis

**DOI:** 10.3390/ijms21176290

**Published:** 2020-08-31

**Authors:** Olivia Patsalos, Bethan Dalton, Hubertus Himmerich

**Affiliations:** 1Department of Psychological Medicine, Institute of Psychiatry, Psychology & Neuroscience, King’s College London, London SE5 8AF, UK; olivia.patsalos@kcl.ac.uk (O.P.); bethan.dalton@kcl.ac.uk (B.D.); 2South London and Maudsley NHS Foundation Trust, London SE5 8AZ, UK

**Keywords:** IL-6, tocilizumab, siltuximab, weight, BMI, spondyloarthritis, rheumatoid arthritis, obesity

## Abstract

Inhibitors of the IL-6 signaling pathway, such as tocilizumab, are frequently administered for the treatment of immune diseases, e.g., rheumatoid arthritis and multicentric Castleman’s disease. The aim of this systematic review and meta-analysis was to ascertain the effects of IL-6 pathway inhibitors on weight and body mass index (BMI). Using PRISMA guidelines, we systematically reviewed relevant articles from three databases (PubMed, OVID, EMBASE). A random effects model was used to estimate standardized mean change (SMCC). Ten studies with a total of 1531 patients were included in the meta-analysis for weight and ten studies with a total of 1537 patients were included in the BMI meta-analysis. The most commonly administered IL-6 pathway inhibitor was tocilizumab. IL-6 pathway inhibitors were associated with increases in weight (SMCC = 0.09, *p* = 0.016, 95% CI [0.03, 0.14]) and BMI (SMCC = 0.10, *p* = 0.0001, 95% CI [0.05, 0.15]). These findings suggest that the IL-6 pathway is involved in weight regulation. Modulating IL-6 signaling may be a potential future therapeutic avenue used as an adjunct for the treatment of disorders associated with weight changes, such as cancer cachexia and anorexia nervosa.

## 1. Introduction

Interleukin (IL)-6 is a small signaling molecule involved in inflammatory processes, initiating fever and mediating the acute phase response. It is a pleiotropic cytokine secreted by a range of cells, such as T cells, B cells, macrophages, osteoblasts, smooth muscle cells and several tumor cells. It is also released by cells in the brain, such as neurons, microglia and astrocytes. IL-6 has been implicated in inflammatory and auto-immune processes related to a wide range of diseases such as diabetes [1], atherosclerosis [2], depression [3], rheumatoid arthritis [4] and Castleman’s disease [5], as well as several cancers [6,7,8] and cancer cachexia [9,10]. Given IL-6’s implication in disease states, several agents inhibiting the IL-6 signaling pathway have been developed as pharmacological treatments for some of these diseases, e.g., rheumatoid arthritis, Castleman’s disease and prostate cancer. Inhibiting the action of IL-6 can be achieved either by targeting the soluble IL-6 molecule or by blocking the cell surface receptor (IL-6R) on which IL-6 attaches. The latter are known as IL-6 receptor antagonists, and the first FDA-approved medication targeting IL-6 signaling, tocilizumab, belongs to this class. Another monoclonal antibody against IL-6R is sarilumab, which has been recently approved by the FDA for the treatment of rheumatoid arthritis. Conversely, siltuximab directly targets the IL-6 soluble molecule and there are several other agents (olokizumab, elsilimomab, sirukumab) in this class that are currently being tested and are at various phases of clinical trials. Even though these agents inhibit the IL-6 signaling pathway via different mechanisms, we have included all IL-6 pathway inhibitors in this review and meta-analysis.

Aside from its involvement in disease states, IL-6 has been shown to play a role in the control of body weight and body fat. For example, in animal studies, overexpression of IL-6 has been reported to reduce fat accumulation and weight gain [11,12,13], whereas IL-6-deficient mice develop mature-onset obesity [14,15]. Previous research points to a complex relationship between IL-6 and body weight and fat. It is generally accepted that high levels of inflammatory mediators such as IL-6 and TNF-α are cachectogenic [9,16,17] and perhaps anorexigenic [18,19], i.e., associated with weight loss and decreased food intake. However, in the context of obesity, low-grade elevation of inflammatory markers is observed, as both aforementioned cytokines are produced by adipose tissue [20,21,22]. Therefore, it is unclear how IL-6 signaling inhibitors may influence weight and/or fat mass.

Generally, administration of IL-6 pathway inhibitors has been associated with an increase in body weight and body mass index (BMI) [23,24], implying that IL-6 signaling inhibition might lead to weight gain as a side-effect. Given the known health risks associated with obesity, it would be of value for clinicians to have reliable information about whether IL-6 signaling inhibitors systematically induce weight gain, in order to aid their decision-making. Additionally, if IL-6 pathway inhibitors induce weight gain, blocking IL-6 signaling could become a novel adjunct strategy for the treatment of disorders with severely low body weight such as cancer cachexia and anorexia nervosa. To ascertain whether IL-6 pathway inhibitors are associated with changes in weight and/or BMI, we performed a systematic review and meta-analysis of the available literature which, to the best of our knowledge, has not yet been collated.

## 2. Results

### 2.1. Characteristics of Included Studies

Individual study characteristics are described in Table 1. Eleven longitudinal studies met the inclusion criteria. Nine studies reported data for both BMI and weight [23,24,25,26,27,28,29,30,31], while one only reported on weight [32] and one only on BMI [33]. Hence, each meta-analysis for weight and BMI included 10 studies. A total of 1537 participants received treatment with an IL-6 pathway inhibitor, with sample sizes ranging from 13 to 805 participants. None of the studies reported correlational data and were thus imputed from a previous meta-analysis by our group [34].

In the included studies, IL-6 signaling pathway inhibitors were administered for the following diseases: nine studies in rheumatoid arthritis patients, one in Castleman’s disease patients and one in obese patients. Gender was reported in eight studies with a total of 952 females and 240 males. Three studies did not report gender. All studies used the IL-6R antagonist tocilizumab, aside from one [28] which used a biosimilar agent called MRA, and was described in the study as a ‘humanized anti-human IL-6 receptor monoclonal antibody of immunoglobulin G1k subtype’. Given its action on IL-6R, the study was deemed appropriate and kept in the review.

Mean age of patients was reported in eight studies and two studies reported a median age. Illness duration was reported in eight studies, with a pooled mean of 8 years. All studies explicitly stated the duration of treatment with an IL-6 signaling pathway inhibitor [23,24,25,26,27,28,29,30,32,33,35]. The shortest treatment duration was 12 weeks [32] and the longest was 60 weeks [28]. The remaining studies reported a range of treatment days depending on adverse events and other reasons for treatment discontinuation. Of the 11 studies included, only five reported the use of additional medication. Those were: methotrexate [25,27,29,30], steroids [30], prednisone [25,27,29] and unspecified DMARD [23]. Smoking status was only reported in one study [25].

### 2.2. Study Findings

#### 2.2.1. Effect of IL-6 Inhibitors on Weight

A meta-analysis of nine studies (one study was removed as it was deemed to be an outlier using Cook’s distance [28]) revealed that patients’ weight was significantly increased after IL-6 signalling pathway inhibitor commencement (SMCC = 0.09, z = 3.16, *p* = 0016, 95% CI [0.03, 0.14]; see Figure 1). The significant between study heterogeneity (I^2^ = 4.06%, Q = 16.20, *p* = 0.04) was further explored using meta-regressions. The meta-regression explained all heterogeneity (Q_moderators_ = 12.91, *p* = 0.0048), leaving no significant, unexplained residual heterogeneity (Q_residual_ = 2.57, *p* = 0.46). The following moderators were included in the final model: diagnosis, time to follow-up, gender and age. The main drivers of between study heterogeneity were a diagnosis of rheumatoid arthritis and age, such that younger patients with rheumatoid arthritis gained more weight. No significant publication bias was exposed by Begg’s rank correlation for funnel plot asymmetry (τ = 1.73, *p* = 0.08).

#### 2.2.2. Effect of IL-6 Signaling Pathway Inhibitors on BMI

Nine studies were subjected to a BMI meta-analysis (one study was removed as it was shown to be an influential outlier using Cook’s distance [28]), which revealed that patients’ BMI was significantly increased at follow-up after IL-6 signaling pathway inhibitor commencement (SMCC = 0.10, z = 3.86, *p* = 0001, 95% CI [0.049, 0.15]; see Figure 2). There was no significance between study heterogeneity (I^2^ = 0%, Q = 8.87, *p* = 0.35). Pooling the mean BMIs of these studies gave a mean baseline BMI of 26.4 kg/m^2^ and a mean post-treatment BMI of 27.1 kg/m^2^. Significant publication bias was exposed by Begg’s rank correlation for funnel plot asymmetry (τ = 2.15, *p* = 0.03).

## 3. Discussion

### 3.1. Summary of the Main Findings

This systematic review and meta-analysis summarize the existing data on the effects of IL-6 signaling pathway inhibitors on weight and BMI. The results from the meta-analysis show that IL-6 pathway inhibitors were associated with increases in weight and BMI. This pattern of weight gain during treatment with an IL-6 pathway inhibitor is in line with research implicating elevated concentrations of IL-6 in the development of cachexia as seen in clinical populations [9,36,37,38,39]. However, it must be considered that, particularly in the case of rheumatoid arthritis where some patients experience weight loss, a restoration of normal body weight may be due to an improvement in disease activity and a reduction in inflammation, rather than a direct effect of the IL-6 signaling pathway inhibitors.

### 3.2. Possible Mechanisms of IL-6-Induced Weight Loss

IL-6 is a functionally pleiotropic cytokine implicated in inflammation and infection responses as well as the regulation of metabolic and neural processes. It has many cell-type specific effects and although primarily regarded as a pro-inflammatory cytokine, IL-6 also has many regenerative or anti-inflammatory properties. Given its wide variety of actions IL-6 has been implicated in many aspects of (patho)physiology, including weight and/or fat mass changes. Research thus far points towards a dual role of IL-6 in the central nervous system (CNS) and the periphery.

#### 3.2.1. Effects on Appetite

With regards to IL-6’s effects on the CNS, there is some evidence indicating that IL-6 might lead to weight loss through a reduction in food intake and/or appetite suppression. For example, in animal studies, where IL-6 was administered intracerebroventricularly, it led to a suppression of food intake, whereas when IL-6 was administered at the same dose intraperitoneally there was no effect on food intake [40,41]. Mishra et al. [41] have postulated that IL-6 exerts its anorexigenic effects through interaction with leptin. Another possible mechanism by which IL-6 could be exerting food intake/appetite control is through its effects on hypothalamic neuropeptides such as neuropeptide Y, agouti-related peptide, melanin-corticotrophin-releasing hormone and pro-opiomelanocortin [13]. With regards to studies in humans, the effect of IL-6 on appetite has been reported by some authors. For example, Hunschede et al. [42] found elevated levels of IL-6 following high intensity exercise in normal weight and obese boys, which was inversely correlated with appetite and fullness. Furthermore, Emille et al. [43] suggested that the observed clinical improvement seen in seven of the eight patients completing treatment with an IL-6 signaling pathway inhibitor for their lymphoma, was due to increased appetite. In a second study by Hunschede et al. [44], it was found that IL-6 correlated with active ghrelin and cortisol, and hence the authors hypothesized that the effects of IL-6 on appetite are potentially mediated by ghrelin and cortisol signaling. In contrast, Li et al. [11] and Hidalgo et al. [12] who centrally expressed murine IL-6 in the hypothalamus of rats found that although weight gain and visceral adiposity was suppressed, food intake was not affected. Thus, research to this date has not consistently confirmed the potential anorexigenic role of IL-6 in the CNS.

#### 3.2.2. Effects on Metabolism

Given the conflicting results of IL-6 effects on food intake and appetite control, it is plausible that other mechanisms are at least partially at play. Alterations in fat mass and body weight without changes in food intake suggest that IL-6 may affect energy metabolism. One of the hallmarks of obesity-related metabolic disorders is chronic, low-grade inflammation. This systemic overabundance of proinflammatory cytokines in adipose tissue, including IL-6, activates STAT3 and subsequently AMPK [45,46] leading to alterations in insulin signaling and eventually Type 2 diabetes [47,48].

On the other hand, there is increasing evidence suggesting a beneficial role for IL-6 in the prevention of obesity and insulin resistance. For example, Carey at al. found that IL-6 increases glucose disposal in healthy humans [45,49]. Additionally, IL-6 deficient mice develop mature-onset obesity, partially attributed to reduced energy expenditure [15], and conversely, overexpression of IL-6 leads to increased thermogenesis [41] energy expenditure [11].

The longstanding assertion that IL-6 leads to insulin resistance has been challenged by the discovery of IL-6’s actions in muscle. Pedersen et al. found that IL-6 acts as a myokine, meaning it is produced and released by muscle cells in response to muscular contractions [50,51], and physical exercise is known to increase insulin sensitivity [52,53]. While there are conflicting findings, it is clear that IL-6 is implicated in the regulation of energy homeostasis, and it is thus possible that IL-6 signaling inhibitors may contribute to changes in weight and fat mass.

### 3.3. Clinical Implications

Our findings suggest that administration of IL-6 signaling pathway inhibitors could lead to weight gain. In instances where obesity is considered a risk factor and an exacerbator of disease, the additional weight gain due to anti-IL-6 administration may be an undesirable effect, and given the well-documented detrimental effects associated with obesity (e.g., cardiovascular disease, metabolic syndrome, mental health problems [54,55,56]), prescribers should take this into consideration when administering IL-6 pathway inhibitors. For such cases, clinical monitoring of patients’ weight with additional weight-regulating measures, such as diet counselling and physical exercise, may be necessary. It is important to mention here that weight gain may not be specific to IL-6 signaling inhibition as other cytokine inhibitors have also been shown to increase weight and BMI [34].

Pharmacological modulation of IL-6 signaling through IL-6 blockade may be a potential route of action for diseases or disorders with a significant weight loss component, such as cancer cachexia and anorexia nervosa. For example, we previously reported elevated levels of IL-6 in anorexia nervosa patients [19], suggesting modulating cytokines such as IL-6 could be a possible treatment option for patients with anorexia nervosa [57,58]. However, IL-6 blockade is associated with other serious side-effects and the clinical use of such medication for the purpose of weight gain should be carefully considered, as the expected weight gain may not justify the burden of additional drug effects.

### 3.4. Strengths and Limitations

This is the first review to systematically collate data and formally explore the effects of IL-6 signaling pathway inhibitors on body weight and BMI, the results of which may have important clinical implications, as discussed above. However, only a small number of studies were available for inclusion in this analysis, and even though there were significant changes in weight and BMI between baseline and follow-up, the effect size was relatively small. Therefore, these results should be interpreted with caution. Furthermore, potentially informative studies had to be excluded as the required data were not available.

In relation to the included studies, an important methodological consideration is that confounding variables were not consistently reported. For example, smoking, a known appetite suppressant, may have influenced body weight in addition to the prescribed IL-6 signaling pathway inhibitor. Additionally, it has been established that other biological agents, such as TNF-α inhibitors, are also associated with weight gain [34] and weight increase might be a common side effect of pro-inflammatory cytokine blockers more generally. Therefore, the weight increase reported here is not specific for IL-6 pathway inhibition. The weight gain identified in our review and meta-analysis may be accounted for by other drugs taken by participants of the included studies. Examples of such drugs are β-receptor antagonists, antipsychotic drugs, and corticosteroids [59]. Indeed, in some studies, patients were receiving concomitant medications, such as methotrexate and corticosteroids. Therefore, we recommend that future research consider the aforementioned variables in their study design and analysis, as well as to systematically assess calorie intake and physical activity to provide insight into possible mechanisms for IL-6 signaling pathway inhibitor-associated weight change.

In our systematic review and meta-analysis, we did not distinguish between direct IL-6 blockers (e.g., siltuximab) and IL-6R blockers (e.g., tocilizumab) so that we could gain a broad understanding of the effect of IL-6 signaling pathway inhibition on body weight regulation. However, it needs to be taken into consideration that IL-6R inhibitors block the receptor and consequently lead to more freely available IL-6. Thus, IL-6 itself increases during the treatment with substances like tocilizumab, particularly at first administration of the drug. When more studies on the different IL-6 signaling pathway inhibitors are available, a more nuanced analysis will be possible.

Lastly, it is worth noting that body weight and BMI are crude measures, which are perhaps less informative. Only one [32] of our included studies provided data on body composition, such as lean mass, fat mass, visceral adiposity, etc., and we were thus unable to formally explore the effects of IL-6 signaling pathway inhibitors on body composition in more detail. It would be of clinical benefit for future studies to incorporate such measures in their data collection to ascertain whether the observed weight gain is due to lean or fat mass increases, as increased fat mass is associated with an increased risk of cardiovascular disease, metabolic syndrome and depression.

## 4. Methods

We conducted this systematic review according to the Preferred Reporting Items for Systematic Reviews and Meta-Analyses (PRISMA) guidelines [60].

### 4.1. Literature Search

Three electronic databases (PubMed, OVID and EMBASE) were systematically searched from inception until 25th May 2020 using the following search terms: anti-IL-6, anti-interleukin-6, IL-6 blocker, interleukin-6 blocker, IL-6 inhibitor, interleukin-6 inhibitor, tocilizumab, siltuximab, olokizumab, elsilimomab, sarilumab, sirukimab in combination with weight, fat mass, body mass and BMI. Reference lists of potentially relevant papers and reviews were also scanned for potentially eligible papers.

### 4.2. Eligibility Criteria

Searches were limited to abstracts, studies with adult human participants, and studies written in English. Any study in which an IL-6 signaling pathway inhibitor was used and assessed weight and/or BMI at baseline and at least one follow-up point was eligible for inclusion.

Studies were excluded if they: (a) did not report values for weight/BMI both at pre-treatment and post-treatment, (b) stratified results by BMI or (c) reported the effect of BMI or weight on IL-6 treatment outcome. Authors that reported such data were contacted to request the raw data. If the raw data was provided, those studies were included in the review. Review articles, meta-analyses, case studies, conference proceedings/abstracts, book chapters, and unpublished theses were not included.

### 4.3. Study Selection

Figure 3 depicts the study selection and screening flowchart. Titles and abstracts of publications resulting from the search were imported into EndNote and duplicates were removed. Two independent reviewers (O. P. and B. D.) performed all stages of the search, screening, and evaluation. Titles and abstracts were screened, and irrelevant articles were disregarded. Articles whose abstracts passed the first screen were read in full and assessed for eligibility based on our pre-specified inclusion criteria, described above. The quality of the included studies was assessed using the Quality Assessment Tool for Before–After (pre–post) Studies With No Control Group from the National Heart, Lung and Blood Institute [61].

### 4.4. Data Extraction and Synthesis

An electronic summary table of the data and other pertinent information was created. Information such as sample size, means, standard deviations of weight and/or BMI, and duration of treatment was collected. Additional parameters of interest such as age, gender, medication type, concurrent medication and clinical diagnosis were also included. If the required data was not reported in the publication, corresponding authors were contacted. Thus, in this systematic review and meta-analysis of IL-6 signaling pathway inhibitors, we followed the same approach as in our previous study on TNF-α inhibitors [34], including the depiction of the selection process (Figure 3) and the results (Figure 1 and Figure 2).

### 4.5. Statistical Analysis

All statistical analyses were conducted in R Studio [62] using the “metafor” package [63]. The primary objective was to test for changes in weight and/or BMI before and after commencement of IL-6 signaling pathway inhibitor treatment. Two separate meta-analyses were performed: one for weight and one for BMI.

For imputing missing standard deviations, the average of the available standard deviations was used. None of the studies reported correlations between baseline and follow-up hence the correlation coefficients were imputed based on the average correlation from a similar meta-analysis conducted by our group [34]. To estimate the differences between baseline and follow-up in body weight and BMI we used standardized mean change with change score standardization (SMCC) [64,65]. To check for the presence of influential outliers we used Cook’s distance [66]. Cohen’s *d* [67] was used to calculate effect size estimates and were considered small if ≥0.20, medium if ≥0.50 and large if ≥0.80. Both within-group variability and between-study heterogeneity was tested through a random effects model using the rma function.

Cochran’s Q and I^2^ were the indices used for between-study heterogeneity. We used forest plots as a means of visualization. To explore whether other variables (such as age, gender, treatment duration, etc.) were responsible for any observed heterogeneity, meta-regressions were employed. Publication bias was assessed using visual inspection of funnel plots and Begg’s rank correlation of funnel plot asymmetry [68].

## 5. Conclusions

The findings from the current review and meta-analysis provide further evidence for a role of IL-6 in body weight regulation. Our results show a small but significant increase in weight and BMI following treatment with an IL-6 pathway inhibitor. This is line with a large body of research performed on animals, as well as in clinical populations, that implicates IL-6 in the control of weight, appetite, and energy metabolism. As weight gain appears to be a side-effect of inhibitors of IL-6 signaling, they may be a potential pharmacological adjunct for the treatment of cancer cachexia and anorexia nervosa. However, given the limited available data and the small average gains, the results of this meta-analysis should be interpreted with caution, and the use of IL-6 pathway blockers should be carefully weighed against their potential side effects.

## Figures and Tables

**Figure 1 ijms-21-06290-f001:**
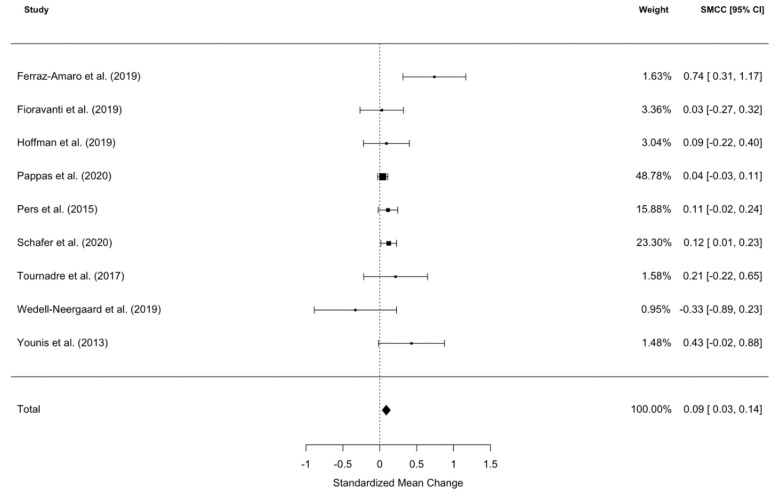
Forest plot of standardized mean change in body weight from nine datasets (*n* = 1531). Zero indicates no effect, whereas points to the right indicate an increase in weight when comparing baseline with follow-up values post-treatment with an IL-6 signaling pathway inhibitor.

**Figure 2 ijms-21-06290-f002:**
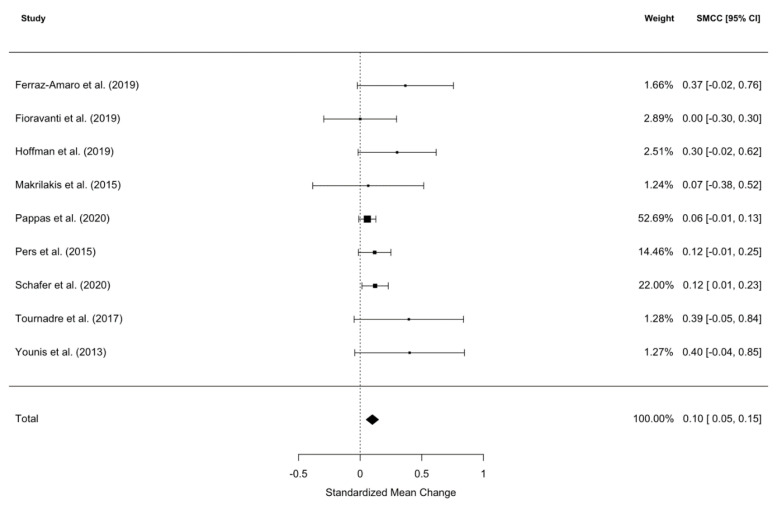
Forest plot of standardized mean change in body mass index (BMI) from nine datasets (*n* = 1537). Zero indicates no effect, whereas points to the right indicate an increase in weight when comparing values at baseline and after treatment with an IL-6 signaling pathway inhibitor.

**Figure 3 ijms-21-06290-f003:**
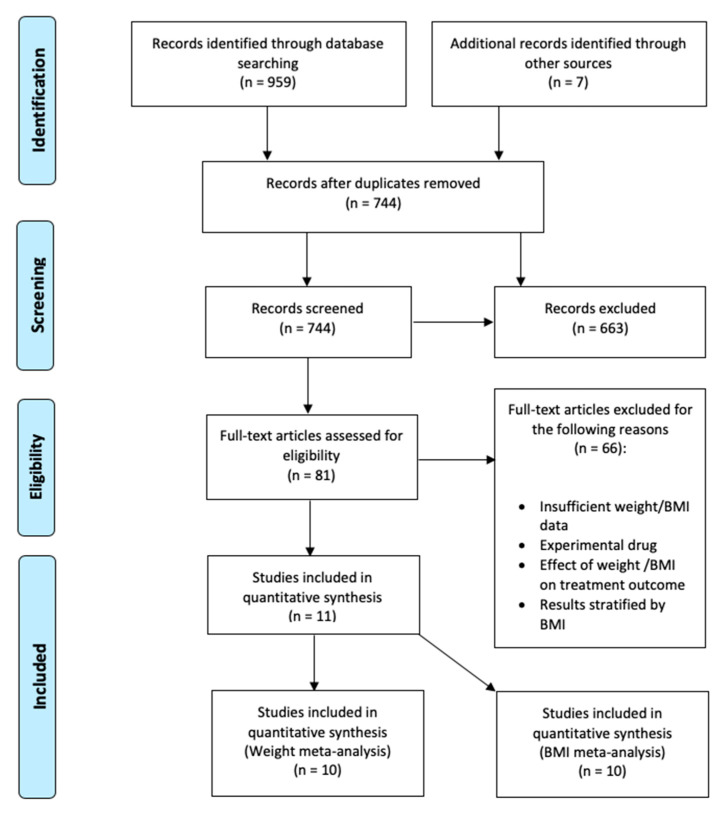
PRISMA flow diagram.

**Table 1 ijms-21-06290-t001:** Study characteristics of included studies.

Study	Disease	Sample Size	Medication	Treatment Duration	Gender F/M	Age (mean ± SD) *Age (Median, 25% and 75% Percentile)*	Concurrent Medication	Summary	Quality Assessment
									Fair
Ferraz-Amaro et al. [25]	RA	27	Tocilizumab	52 weeks		52 ± 11	Methotrexate Prednisone	BMI ↔	Fair
Fioravanti et al. [26]	RA	44	Tocilizumab	26 weeks	38/6	*58.5 (48–69.8)*		Weight ↔ BMI ↔	Fair
Hoffman et al. [27]	RA	40	Tocilizumab	16 weeks	33/7	57.5 ± 11.1	Methotrexate Prednisone	Weight ↔ BMI ↔	Fair
Makrilakis et al. [33]	RA	19	Tocilizumab	26 weeks	18/1	48.6 ± 10.9		BMI ↑	Fair
Nishimoto et al. [28]	CD	28	MRA	60 weeks	11/17	*38*		Weight ↑ BMI ↑	Fair
Pappas et al. [29]	RA	805	Tocilizumab	52 weeks	645/160	58 ± 13	Methotrexate Prednisone	Weight ↔ BMI ↔	Fair
Pers et al. [30]	RA	222	Tocilizumab	26 weeks	183/39	55.5 ± 13.9	Methotrexate Steroids	Weight ↑ BMI ↑	Good
Schäfer et al. [35]	RA	338	Tocilizumab	52 weeks				Weight ↑ BMI ↑	Good
Tournadre et al. [23]	RA	21	Tocilizumab	52 weeks	16/5	57.8 ± 10.5	Unspecified DMARD	Weight ↑ BMI ↑	Fair
Wedell-Neergaard et al. [32]	Obesity	13	Tocilizumab	12 weeks	8/5	44 ± 12		Weight ↔	Good
Younis et al. [24]	RA, SpA	21	Tocilizumab	16 weeks		49.8 ± 14.6		Weight ↑ BMI ↑	Fair

Abbreviations: F = female; M = male; SD = standard deviation; BMI = body mass index; CD = Castleman’s disease; RA = rheumatoid arthritis; SpA = spondyloarthritis; DMARD = disease-modifying anti-rheumatic drugs; ↑ indicates increase (with or without significance testing); ↔ indicates no change (with or without significance testing).

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
