# Peer review of "Effects of IL-6 Signaling Pathway Inhibition on Weight and BMI: A Systematic Review and Meta-Analysis"

_ijms, 2020, doi:10.3390/ijms21176290_

Round 1

Reviewer 1 Report

The authors adreesed adequately my previous comments. Please include the weight (%) of each one of the studies in the figures of the meta-analysis

Author Response

We thank reviewer 1 for their comments. We have included the weight (%) in figures 1 and 2.

Reviewer 2 Report

The authors addressed all my commentaries.

Author Response

We thank reviewer 2 for reviewing our manuscript a second time and for their positive evaluation of our article.

This manuscript is a resubmission of an earlier submission. The following is a list of the peer review reports and author responses from that submission.

Round 1

Reviewer 1 Report

This is a systematic review on the effets of IL-6 inhibitors in different diseases, mainly rheumatoid arthritis. Although the study is interesting I have some important concerns about  the methodology used and the interpretation of the results

  1. This is a systematic review of 14 studies (that finally met the inclusion criteria) of patients treated with IL-6 inhibitors (in 10 studies tocilizumab). The main objective is to analyse the weight changes after treatment with only one study analyzing the effect on fat mass. The conclusion, according to the results of the different studies is that IL-6 inhibition increases the body weight and several interesting possible mechanisms have been elucidated. My major concern is that not all studies confirm this weight increase and the magnitude of the effect in some of them is quite low. Why the authors do not perform a metaanalysis after the systematic review.?. It would be more clarifying in my opinion
  2. The authors suggest causality between weight increase and IL-6 inhibition and this is difficult to confirm in these type of studies, especially in those including patients with RA. It is well demonstrated that other biological agents, such TNF inhibitors may increase also body weight in patients with RA or other immunomediated diseases; a recent metaanalysis of the same group (Front Pharmacol 2020) confirms this statement. Therefore the weight increase is not specific for IL-6 inhibition and may account for other drugs. This should be mentioned in the discussion section. On the other hand the weight changes in these patients may be due to an improvement of disease activity with a reduction of inflammation (good response to therapy)  rather to a direct drug effect, at least in patients with RA.
  3. One of the limitations of the study is the confounding variables as it has been addressed by the authors specially concomitant medication. The authors pointed out the possible effect of methotrexate but in my opinion the effects of steroids are more relevant¡¡¡

Reviewer 2 Report

In this systematic review, Patsalos, et al. have analysed the effect of IL-6 inhibitors on weight and adiposity. Different cytokines such as IL-6 or TNFalpha contributes to spread the pro-inflammatory enviroment. Obesity and metabolic diseases has been partly considered inflammatory conditions. Therefore, blocking these cytokines would have an effect on weight regulation. Herein, authors suggest that IL-6 inhibition results in increased BMI.

Although this systematic review seems extensive and accurate, some suggestions may be improve the manuscript:

1) Although, tocilizumab or sarilumab are considered IL-6 inhibitors, this is completely wrong. Biopharmaceuticals as tocilizumab or sarilumab are IL-6R (IL-6 receptor) blockers. Therefore, these drugs block the receptor and consequently leaves IL6 free. So, this should be taken into account along the manuscript to allow a proper discussion of the results. In fact, IL-6 increases during the treatment with tocilizumab, particularly at the first administrations of the drug. 

2) I would encourage the authors to make a meta-analysis in order to identify the overall degree of the effect of IL6 pathway inhibition on the body weight (BMI). This would be allow readers a quick perspective of the phenomenon.